# Modeling and Analyzing Preemption-Based Service Prioritization in 5G Networks Slicing Framework

Yves Adou [1,*] , Ekaterina Markova [1] and Yuliya Gaidamaka [1,2]

1   Applied Probability and Informatics Department, Peoples' Friendship University of Russia
    (RUDN University), 6 Miklukho-Maklaya Str., Moscow 117198, Russia;
2   Federal Research Center "Computer Science and Control" of the Russian Academy of Sciences,
    44-2 Vavilov Str., Moscow 119333, Russia
*   Correspondence: 1042205051@rudn.ru

**Abstract:** The Network Slicing (NS) technology, recognized as one of the key enabling features of Fifth Generation (5G) wireless systems, provides very flexible ways to efficiently accommodate common physical infrastructures, e.g., Base Station (BS), multiple logical networks referred to as Network Slice Instances (NSIs). To ensure the required Quality of Service (QoS) levels, the NS-technology relies on classical Resource Reservation (RR) or Service Prioritization schemes. Thus, the current paper aims to propose a Preemption-based Prioritization (PP) scheme "merging" the classical RR and Service Prioritization schemes. The proposed PP-scheme efficiency is evaluated or estimated given a Queueing system (QS) model analyzing the operation of multiple NSIs with various requirements at common 5G BSs. As a key result, the proposed PP-scheme can provide up to 100% gain in terms of blocking probabilities of arriving requests with respect to some baseline.

**Keywords:** 5G; slicing; priority; pre-emption; service; isolation; GBR; requirement; queueing; resource





## 1. Introduction

With the emergence of new opportunities in the era of 5G and Cloud technologies, the industries, services and users have various requirements for the Quality of Service (QoS). For example, Mobile communications, Environmental monitoring, Smart Home, Smart Agriculture, and Smart Metering require a huge number of connected devices and frequent transmission of numerous small packets. Other examples are Live Streaming, Video Uploads, and Mobile Health services that require higher data rates, as well as the Internet of Vehicles (IoV), Smart Grids, and industrial Internet of Things (IIoT) services requiring millisecond latency and near 100% reliability. As documented by standards organizations, namely the 3rd Generation Partnership Project (3GPP), GSM Association, and others, 5G networks should provide features such as mass access, deterministic latency, and ultra-high reliability. In this regard, it has become necessary to create flexible and dynamic networks that meet the various requirements for servicing users and vertical industries [1–5].

To flexibly meet the various requirements of users, the Network Slicing (NS) technology is introduced [1–11]. Given NS-technology, Mobile Network Operators (MNOs) can enable multiple dedicated, virtualized, and isolated networks known as Network Slice Instances (NSIs) at common physical network infrastructures, e.g., Base Station (BS). Once fully established, interface specifications for Transport Network (TN), Core Network (CN), and Data Network (DN) control domains enable the automatic creation and deletion of 5G NSIs, and allow MNOs to monitor QoS in networks for smooth Degradation detection [5,12–15]. NSI can only be designed or instantiated to accommodate a unique service type from the standardized categories [1,2,16–18], i.e., guaranteed bit rate (GB), best effort with minimum guaranteed (BG), and best effort (BE). Note that GB category regroups the services that require minimum and maximum bit rates, e.g., Video Streaming, Music

Streaming, etc. As for BG category, it regroups the services that only require minimum bit rate, e.g., Social Networking, Web Surfing or Browsing, etc. Lastly, BE category regroups the services that do not require minimum and/or maximum bit rates, e.g., Smart Metering, Email, etc.

The main objective of NS-technology is the efficient resource use while avoiding network downtimes and ensuring the required NSIs isolation [14,19–21]. In that way, load bursts in some NSIs cannot affect the QoS in other neighboring NSIs. The outlined by Kleinrock classical approach to finite Resource Sharing introduces five Resource Allocation schemes from Complete Partitioning to Complete Sharing through schemes with Maximum Queue Length and Minimum Allocation [22,23]. This approach, based on the Resource Reservation (RR) scheme, cannot allow reserved resources use even when network downtimes occur. Thus, the approach cannot ensure the efficient resource use in 5G systems. Moreover, the Service Prioritization scheme required in 5G networks for heterogeneous traffic service is not considered in the approach.

In 5G networks, the RR and Service Prioritization schemes can contribute to NSIs isolation [24–29]. Given the traffic intensities, the RR-scheme is relatively efficient and simple to implement, providing guaranteed minimum and/or maximum data rates to NSIs at common 5G BSs. The drawback is that this scheme can prove inefficient and limit the network flexibility to accommodate any new demand. In contrast, inherently dynamic and allowing efficient resource use, the Service Prioritization can pave the way to a solution, though hierarchy creation among the NSIs cannot enforce the required isolation.

Given what precedes, the paper [26] focuses on three important features of NS-technology: flexible priority-based performance isolation, fair QoS-aware resource allocation among users and efficient resource use. A Queueing System (QS) model to analyze the operation of three NSIs at common 5G BSs is proposed. The NSIs are considered to each support a unique service type from the standardized GB, BG, and BE categories. The authors assumed uniform data rate requirements for starting service of users at the NSIs. Thus, is proposed for heterogeneous traffic service a NS scheme that can bolster the NSIs performance isolation while maintaining efficient resource use at the Complete Sharing level. In addition, the paper [25] proposes a QS model to estimate the End-to-End (E2E) mean packet delay of the NSIs. The 5G infrastructure performance in industrial environments is analyzed with a focus on the instantiated NSIs isolation degree. A reasonably complete and realistic setup is considered, given inputs parameters from experimentation and simulation. The aim is to highlight the features enabled when using segregated NSIs with reserved resources to service the traffic generated by various production lines at a factory floor. As an important feature to solution QoS degradation, the number of production downtimes and the corresponding associated expenditures can be significantly reduced. In contrast, the paper [27] proposes a real test-bed of 5G stand-alone network deployment with a Service Prioritization scheme, then a NS-technology technique, and, finally, a flexible transparent allocation mechanism at the Radio level. The first aim is to mark a difference with other contributions remaining at the simulation level. The proposed approaches are evaluated in terms of performance given a scenario with saturated uplinks communications. Multiple data flows and various service requirements from external sources are considered in the contribution. The results obtained are then compared to others, from a scenario without NS-technology.

In the current article, following the papers [27,30] and in contrast with papers [25,26], we focus on two key features of NS-technology: NSIs Resource isolation and Automated management. First, propose a Pre-emption-based Prioritization (PP) scheme "merging" the classical RR and Service Prioritization schemes. Second, and last, evaluate or estimate the proposed PP-scheme efficiency given a QS model analyzing the operation of multiple NSIs with various requirements at common 5G BSs. Thus, the remainder of this paper is outlined as follows. Section 2 discusses the main assumptions and considerations of the paper. Section 3 mathematically analyzes a given QS model and proposes formulas to find

key performance indicators (KPIs). Section 4 gives numerical results, illustrative of the proposed PP-scheme efficiency. Section 5 presents the conclusions.

## 2. System Model

Consider a Fifth Generation (5G) Base Station (BS) at which operate customizable and logical Network Slice Instances (NSIs). Let $C$ represent the total network capacity of the 5G BS measured in some capacity units. For simplicity, assume that constant network capacities and service requirements for a NSI are measured in bps, i.e., these can be given in other capacity units by applying the radio channel model for the Radio Access Technology (RAT) in use. Let $\mathcal{S}$ represent the finite set of NSIs at the 5G BS, i.e., $\mathcal{S} \subset \mathbb{N} \setminus \{0\}$ with $S = |\mathcal{S}|$ representing the number of NSIs. Let $C_s$ represent the overall network capacity of the $s^{\text{th}}$ NSI, $s \in \mathcal{S}$, under the condition

$$C_1 + \ldots + C_S \geq C. \tag{1}$$

Suppose that the overall network capacity $C_s$ of the $s^{\text{th}}$ NSI includes a guaranteed network capacity [26,31]. Thus, let $Q_s$ represent the guaranteed network capacity of the $s^{\text{th}}$ NSI, i.e., $Q_s \leq C_s$, under the condition

$$Q_1 + \ldots + Q_S \leq C. \tag{2}$$

The above conditions suggest that when no requests are servicing at the $s^{\text{th}}$ NSI, the guaranteed network capacity $Q_s$ is available to service arriving requests at the other NSIs. In that case, these arriving requests become violators of the isolation of the $s^{\text{th}}$ NSI. At those moments, arriving requests at the $s^{\text{th}}$ NSI become competent to free the necessary resources from the violators servicing at the other NSIs. Various approaches are used to determine the number and the emplacement of the violators that must be discarded to free resources [32]. Some approaches suggest discarding the last admitted violator, and others propose discarding the violator with the longest remaining service time. Our paper proposes a Pre-emption-based Prioritization (PP) scheme for randomly discarding first the violators servicing at the NSI with the lowest priority until the necessary resources are freed up.

Consider the Poisson arrival process of a unique request type with rate $\lambda_s$, $s \in \mathcal{S}$, at the $s^{\text{th}}$ NSI. Suppose that the arriving request at the $s^{\text{th}}$ NSI requires for starting service $b_s$ capacity units, i.e., $b_s \leq Q_s$. Assume the average service time for a request at the $s^{\text{th}}$ NSI to be exponentially distributed, with the mean $\mu_s^{-1}$ that corresponds to scenarios with real-time applications.

Give a summary of the proposed system main notations in Table 1.

Organize the radio admission control (RAC) scheme so that upon arrival at the $s^{\text{th}}$ NSI, $s \in \mathcal{S}$, the arriving request is bound to one path: blocking, direct admission, or via pre-emption admission. The blocking path is followed when the number $n_s$ of servicing requests at the NSI is greater than or equal to $\lfloor Q_s / b_s \rfloor$ and the amount of available capacity units at the 5G BS is less than $b_s$. The direct admission path is followed when the number $n_s$ of servicing requests at the NSI is less than $\lfloor C_s / b_s \rfloor$ and the amount of available capacity units at the 5G BS is greater than or equal to $b_s$. The via pre-emption admission path is followed when the number $n_s$ of servicing requests at the NSI is less than $\lfloor Q_s / b_s \rfloor$ and the amount of available capacity units at the 5G BS is less than $b_s$. In that case, the arriving request is competent to discard a number of violators servicing at the $\hat{s}^{\text{th}}$ NSI, $\hat{s} \in \mathcal{S} \setminus \{s\}$. Consideration of Service Prioritization appears with the dilemma of choosing the violator(s) that must be discarded in the cases of three and more NSIs at the 5G BS. Such cases are outlined below. Therefore, develop a Discard or Preemption scheme, starting with cases of the system where two and three NSIs operate.

**Table 1.** System parameters.

| Notation | Description |
|---|---|
| $\mathcal{S}$ | The set of NSIs at the 5G BS, $\mathcal{S} \subset \mathbb{N} \setminus \{0\}$, [units (u.)] |
| $S$ | The number of NSIs at the 5G BS, $S = |\mathcal{S}|$, [u.] |
| $C$ | The total network capacity of the 5G BS, [capacity units (c.u.)] |
| $C_s$ | The overall network capacity of the $s^{\text{th}}$ NSI, $s \in \mathcal{S}$, $C_1 + \ldots + C_S \geq C$, [c.u.] |
| $Q_s$ | The guaranteed network capacity of the $s^{\text{th}}$ NSI, $Q_s \leq C_s$, $Q_1 + \ldots + Q_S \leq C$, [c.u.] |
| $\lambda_s$ | The arrival rate of requests at the $s^{\text{th}}$ NSI, $\boldsymbol{\lambda} = (\lambda_1, \ldots, \lambda_S)$, [requests per time units (requests/t.u.)] |
| $\mu_s^{-1}$ | The average service time for a request at the $s^{\text{th}}$ NSI, $\boldsymbol{\mu} = (\mu_1, \ldots, \mu_S)$, [t.u.] |
| $\rho_s = \lambda_s / \mu_s$ | The offered load at the $s^{\text{th}}$ NSI |
| $b_s$ | The requirement for starting service of a request at the $s^{\text{th}}$ NSI, $b_s \leq Q_s$, $\mathbf{b} = (b_1, \ldots, b_S)$, [c.u.] |
| $\lfloor C_s / b_s \rfloor$ | The maximum number of requests that may be admitted for service with the overall network capacity of the $s^{\text{th}}$ NSI, $\mathbf{N}^{\text{max}} = (\lfloor C_1 / b_1 \rfloor, \ldots, \lfloor C_S / b_S \rfloor)$, [u.] |
| $\lfloor Q_s / b_s \rfloor$ | The maximum number of requests that may be admitted for service with the guaranteed network capacity of the $s^{\text{th}}$ NSI, $\mathbf{N}^{\text{g}} = (\lfloor Q_1 / b_1 \rfloor, \ldots, \lfloor Q_S / b_S \rfloor)$, [u.] |
| $n_s$ | The current number of servicing requests at the $s^{\text{th}}$ NSI, $\mathbf{n} = (n_1, \ldots, n_S)$, [u.] |
| $\mathbf{e}_s$ | The $s^{\text{th}}$ row of the size $S \times S$ identity matrix |
| $\mathbf{j}$ | The $S$-dimensional all-ones vector |

### 2.1. Case Example of Two NSIs

Consider the case example of two NSIs, i.e., $\mathcal{S} = \{1, 2\}$, instantiated at a common 5G BS. Clarify the Pre-emption scheme by supposing that with an arriving request bound to the via pre-emption admission path at the $s^{\text{th}}$ NSI, i.e., $s \in \{1, 2\}$, a number

$$u_{\hat{s}}^{(s,\mathbf{n})} = \left\lceil \frac{(\mathbf{n} + \mathbf{e}_s) \cdot \mathbf{b} - C}{\mathbf{e}_{\hat{s}} \cdot \mathbf{b}} \right\rceil, \quad \hat{s} \in \{1, 2\} \setminus \{s\}, \tag{3}$$

of violators servicing at the $\hat{s}^{\text{th}}$ NSI undergo a service pre-emption or are discarded.

As an example application, with an arriving request bound to the via pre-emption path at the 1st NSI, exactly $u_2^{(1,\mathbf{n})}$ violators servicing at the 2nd NSI are discarded. Analogically, with an arriving request bound to the via pre-emption path at the 2nd NSI, exactly $u_1^{(2,\mathbf{n})}$ violators servicing at the 1st NSI are discarded.

Thus, improve the Pre-emption scheme for three NSIs in the following subsection.

### 2.2. Case Example of Three NSIs

Consider the case example of three NSIs, i.e., $\mathcal{S} = \{1, 2, 3\}$, instantiated at a common 5G BS. Differently to the previous subsection, introduce Priority levels [3,24,29,33,34] to clarify the Pre-emption scheme. Let

- The highest priority be assigned to all servicing requests at the 1st NSI;
- The medium priority be assigned to all servicing requests at the 2nd NSI;
- The lowest priotity be assigned to all servicing requests at the 3rd NSI.

Thus, consider that with an arriving request bound to the via pre-emption admission path at the $s^{\text{th}}$ NSI, $s \in \{1, 2, 3\}$, the following consecutive events occur at the system. Firstly, a number

$$u_{\hat{s}}^{(s,\mathbf{n})} = \min\left\{ (\mathbf{n} - \mathbf{N}^{\text{g}}) \cdot \mathbf{e}_{\hat{s}}, \left\lceil \frac{(\mathbf{n} + \mathbf{e}_s) \cdot \mathbf{b} - C}{\mathbf{e}_{\hat{s}} \cdot \mathbf{b}} \right\rceil \right\},$$
$$\hat{s} = \max\{i \in \{1, 2, 3\} \setminus \{s\} : (\mathbf{n} - \mathbf{N}^{\text{g}}) \cdot \mathbf{e}_i > 0\}, \tag{4}$$

of violators servicing at the $\hat{s}^{\text{th}}$ NSI are discarded. Secondly and last, a number

$$u_{\tilde{s}}^{(s,\mathbf{n})} = \left\lceil \frac{\left(\mathbf{n} + \mathbf{e}_s - u_{\hat{s}}^{(s,\mathbf{n})} \mathbf{e}_{\hat{s}}\right) \cdot \mathbf{b} - C}{\mathbf{e}_{\tilde{s}} \cdot \mathbf{b}} \right\rceil, \quad \tilde{s} \in \{1, 2, 3\} \setminus \{s, \hat{s}\}, \tag{5}$$

of violators servicing at the $\tilde{s}^{\text{th}}$ NSI are also discarded.

As an example application, with an arriving request bound to the via pre-emption path at the 2nd NSI, exactly $u_3^{(2,\mathbf{n})}$, then $u_1^{(2,\mathbf{n})}$ violators servicing, respectively, at the 3rd and 1st NSIs are discarded.

Thus, generalize the Pre-emption scheme for multiple NSIs in the following subsection.

### 2.3. General Case of S NSIs

Consider the general case of $S$ NSIs, i.e., $\mathcal{S} \subset \mathbb{N} \setminus \{0\}$, instantiated at a common 5G BS. Similarly to previous Subsection, introduce Priority levels to clarify the Pre-emption scheme. Let $s$ represent the priority level of all servicing requests at the $s^{\text{th}}$ NSI, e.g., the highest priority is assigned to all servicing requests at the 1st NSI, and the lowest priority is assigned to all servicing requests at the $S^{\text{th}}$ NSI. Thus, introduce the Pre-emption vector-function

$$\mathbf{u}^{(s,\mathbf{n})} = \left( u_{\hat{s}}^{(s,\mathbf{n})} \right)_{\hat{s}=1,\dots,S} = \left( u_1^{(s,\mathbf{n})}, \dots, u_S^{(s,\mathbf{n})} \right), \quad s \in \mathcal{S}, \tag{6}$$

whose entries $u_{\hat{s}}^{(s,\mathbf{n})}$, $\hat{s} \in \mathcal{S}$, represent the number of violators servicing at the $\hat{s}^{\text{th}}$ NSI that must be discarded when an arriving request is bound to the via pre-emption admission path at the $s^{\text{th}}$ NSI.

**Theorem 1.** *Given the initial condition $\mathbf{u}^{(s,\mathbf{n})} = \mathbf{0}$, describe the entry*

$$u_{\hat{s}}^{(s,\mathbf{n})} = \min\left\{ R\{(\mathbf{n} - \mathbf{N}^{\text{g}}) \cdot \mathbf{e}_{\hat{s}}\}, R\left\{ \left\lceil \frac{\left(\mathbf{n} + \mathbf{e}_s - \mathbf{u}^{(s,\mathbf{n})}\right) \cdot \mathbf{b} - C}{\mathbf{e}_{\hat{s}} \cdot \mathbf{b}} \right\rceil \right\} \right\}, \quad \hat{s} = S, \dots, 1, \tag{7a}$$

*where $R\{x\} = xH(x)$ represents the Ramp function (https://mathworld.wolfram.com/RampFunction.html, accessed 1 September 2022), and $H(x)$—the Heaviside function:*

$$H(x) = \begin{cases} 1, & x > 0, \\ 0, & \text{otherwise.} \end{cases} \tag{7b}$$

Note that when $\hat{s} = s$ the entry $u_{\hat{s}}^{(s,\mathbf{n})}$ is zero, i.e., an arriving request cannot follow the via pre-emption admission path at the expense of one servicing request at the same NSI.

Given the Pre-emption vector-function (Theorem 1), formalize in Figure 1 the RAC scheme for accessing the $s^{\text{th}}$ NSI, i.e., $s \in \mathcal{S}$, at the 5G BS.

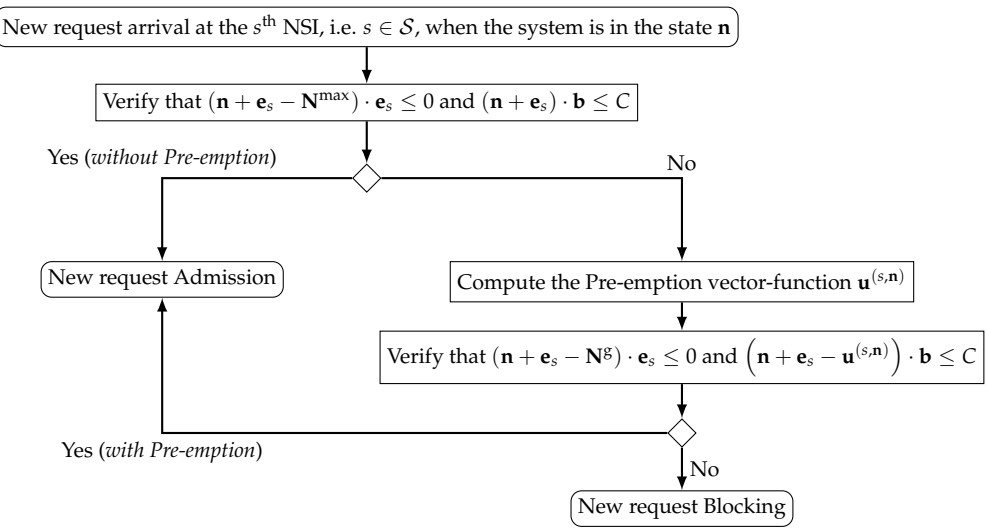

**Figure 1.** Flowchart formalizing the RAC scheme for accessing the $s^{\text{th}}$ NSI, i.e., $s \in \mathcal{S}$, at the 5G BS.

Evaluate and analyze the efficiency of the proposed Pre-emption-based Prioritization (PP) scheme stated by Theorem 1. To this end, construct a mathematical model using the Queueing Theory apparatus, and propose formulas to calculate key performance indicators (KPIs) in the following section.

## 3. Mathematical Model

Given the Poisson arrival processes, the exponentially distributed service times, plus the RAC scheme, describe the system behavior using a *S*-dimensional Markov process

$$\mathbf{X}(t) = (X_1(t), \ldots, X_S(t)), \quad t > 0, \tag{8}$$

where $X_s(t)$, $s \in \mathcal{S}$, represents the number of servicing requests at the $s^{\text{th}}$ NSI at the time $t$ over the system state space

$$\Omega = \left\{ \mathbf{n} \in \mathbb{N}^S : (\mathbf{n} - \mathbf{N}^{\text{max}}) \cdot \mathbf{j} \leq 0 \wedge \mathbf{n} \cdot \mathbf{b} \leq C \right\}, \tag{9}$$

where $\mathbb{N}^S$ represents the set of all *S*-dimensional vectors with natural elements.

A depiction of considered Queueing System (QS) functioning is given by the scheme model in Figure 2.

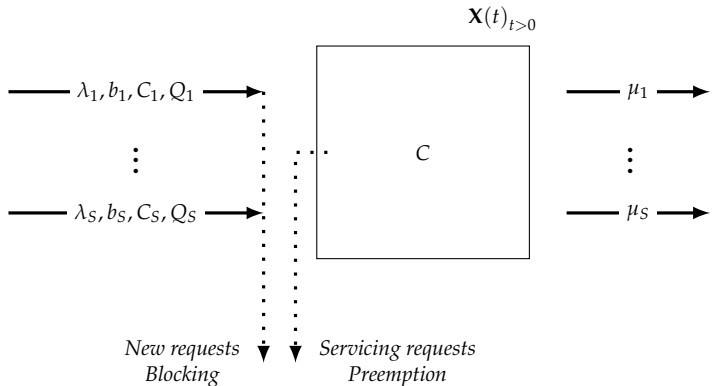

**Figure 2.** Scheme model of considered QS with *S* NSIs at the 5G BS.

Consider the following main sets [35–37] of the state space $\Omega$ for further investigation of the model. Let the blocking set, described as

$$\Omega_s^{\text{block}} = \{ \mathbf{n} \in \Omega : (\mathbf{n} - \mathbf{N}^{\text{g}}) \cdot \mathbf{e}_s \geq 0 \wedge (\mathbf{n} + \mathbf{e}_s) \cdot \mathbf{b} > C \}, \quad s \in \mathcal{S}, \tag{10}$$

collect all the system states, where an arriving request is bound to the blocking path at the $s^{\text{th}}$ NSI. Let the direct admission set, described as

$$\Omega_s^{\text{dad}} = \{ \mathbf{n} \in \Omega : (\mathbf{n} - \mathbf{N}^{\text{max}}) \cdot \mathbf{e}_s < 0 \wedge (\mathbf{n} + \mathbf{e}_s) \cdot \mathbf{b} \leq C \}, \tag{11}$$

collect all the system states, where an arriving request is bound to the direct admission path at the $s^{\text{th}}$ NSI. Let the via pre-emption admission set, described as

$$\Omega_s^{\text{vpad}} = \{ \mathbf{n} \in \Omega : (\mathbf{n} - \mathbf{N}^{\text{g}}) \cdot \mathbf{e}_s < 0 \wedge (\mathbf{n} + \mathbf{e}_s) \cdot \mathbf{b} > C \}, \tag{12}$$

collect all the system states, where an arriving request is bound to the via pre-emption admission path at the $s^{\text{th}}$ NSI. Given the direct admission and via pre-emption admission sets, also describe the blocking set as

$$\Omega_s^{\text{block}} = \Omega \setminus \left( \Omega_s^{\text{dad}} \cup \Omega_s^{\text{vpad}} \right), \tag{13}$$

where $\Omega_s^{\text{dad}} \cup \Omega_s^{\text{vpad}}$ represents the collection of all the system states that are in $\Omega_s^{\text{dad}}$ or in $\Omega_s^{\text{vpad}}$, i.e., an arriving request is bound to any of the two admission paths at the $s^{\text{th}}$ NSI.

A depiction of the system transition diagram, for an arbitrary state $\mathbf{n}$, i.e., $\mathbf{n} \in \Omega$, is given in Figure 3.

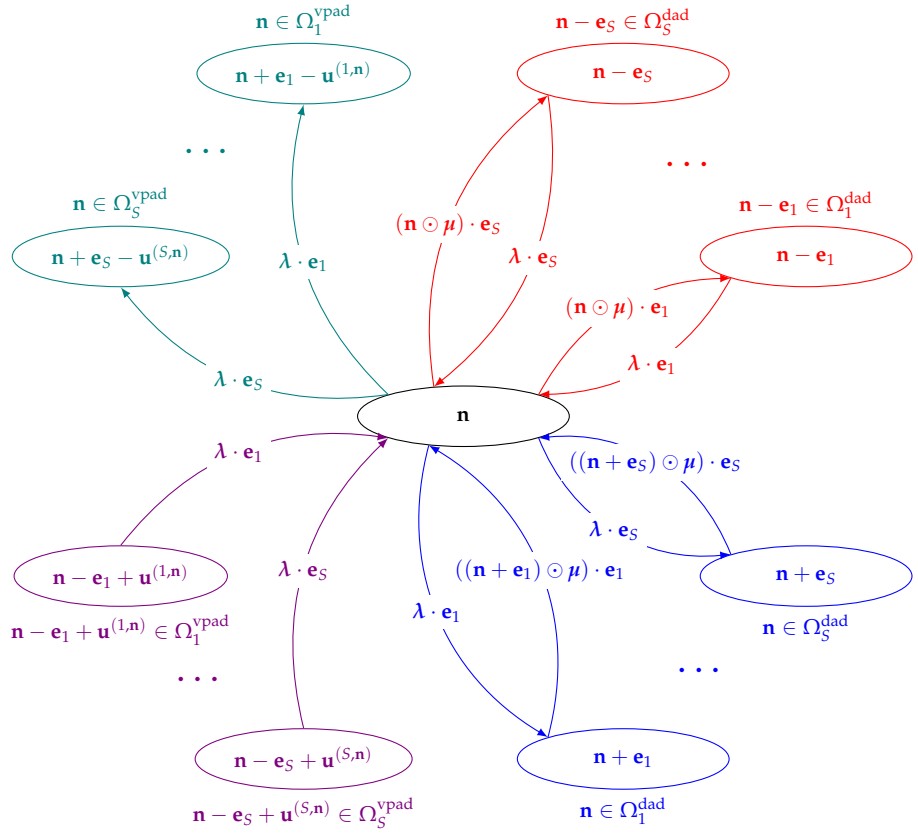

**Figure 3.** Transition diagram fragment for an arbitrary system state $\mathbf{n}$, i.e., $\mathbf{n} \in \Omega$.

Introduce the stationary probabilities of the Markov process $\mathbf{X}(t)$

$$P(\mathbf{n}) = \lim_{t \to \infty} \mathsf{P}\{\mathbf{X}(t) = \mathbf{n}\}, \quad \mathbf{n} \in \Omega,$$

and describe them using the system of equilibrium equations

$$P(\mathbf{n}) \left( \lambda \cdot \sum_{s=1}^{S} \left( I_{\Omega_s^{\text{dad}}}\{\mathbf{n}\} + I_{\Omega_s^{\text{vpad}}}\{\mathbf{n}\} \right) \mathbf{e}_s + \mathbf{n} \cdot \mu \right) =$$

$$= \lambda \cdot \sum_{s=1}^{S} \left( P(\mathbf{n} - \mathbf{e}_s) \, I_{\Omega_s^{\text{dad}}}\{\mathbf{n} - \mathbf{e}_s\} + P\left(\mathbf{n} - \mathbf{e}_s + \mathbf{u}^{(s,\mathbf{n})}\right) I_{\Omega_s^{\text{vpad}}}\left\{\mathbf{n} - \mathbf{e}_s + \mathbf{u}^{(s,\mathbf{n})}\right\} \right) \mathbf{e}_s +$$

$$+ \mu \cdot \sum_{s=1}^{S} \left( P(\mathbf{n} + \mathbf{e}_s) \, I_{\Omega_s^{\text{dad}}}\{\mathbf{n}\} \, (\mathbf{n} + \mathbf{e}_s) \cdot \mathbf{e}_s \right) \mathbf{e}_s, \quad \mathbf{n} \in \Omega, \quad (14\text{a})$$

where $P(\mathbf{n})$ represents the stationary probability that the system is in the state $\mathbf{n}$ and $I_{\mathcal{A}}\{a\}$ represents the Indicator function (https://mathworld.wolfram.com/CharacteristicFunction.html, accessed 1 September 2022):

$$I_{\mathcal{A}}\{a\} = \begin{cases} 1, & a \in \mathcal{A}, \\ 0, & \text{otherwise.} \end{cases} \quad (14\text{b})$$

Note that the Markov process describing the system behavior is not reversible. Therefore, compute the system stationary probability distribution $\mathbf{P} = [P(\mathbf{n})]_{\mathbf{n} \in \Omega}$, i.e., a size $|\Omega| \times 1$ matrix, using an iterative method [38–40] to solve the system of equilibrium equations, rewritten as

$$\mathbf{A}^\top \mathbf{P} = \mathbf{0}, \quad \mathbf{P} \cdot \mathbf{j} = 1, \tag{15}$$

where $\mathbf{A}$ represents the infinitesimal generator of Markov process, i.e., a size $|\Omega| \times |\Omega|$ matrix, whose entries $\mathsf{A}(\mathbf{n} \in \Omega, \hat{\mathbf{n}} \in \Omega)$ are computed as follows:
when $\mathbf{n} \neq \hat{\mathbf{n}}$,

$$\mathsf{A}(\mathbf{n}, \hat{\mathbf{n}}) = \begin{cases} \boldsymbol{\lambda} \cdot \mathbf{e}_s, \text{ if } \hat{\mathbf{n}} = \mathbf{n} + \mathbf{e}_s, & \text{s.t. } \mathbf{n} \in \Omega_s^{\mathrm{dad}}, \\ \quad \text{elseif } \hat{\mathbf{n}} = \mathbf{n} + \mathbf{e}_s - \mathbf{u}^{(s,\mathbf{n})}, & \text{s.t. } \mathbf{n} \in \Omega_s^{\mathrm{vpad}}, \\ (\mathbf{n} \odot \boldsymbol{\mu}) \cdot \mathbf{e}_s, \text{ if } \hat{\mathbf{n}} = \mathbf{n} - \mathbf{e}_s, & \text{s.t. } \hat{\mathbf{n}} \in \Omega_s^{\mathrm{dad}}, \\ \quad 0, \text{ otherwise}, & \text{i.e., } \hat{\mathbf{n}} \in \Omega \setminus \{\mathbf{n}\}, \end{cases}$$

$$\text{i.e., } s = 1, \dots, S, \tag{16a}$$

when $\mathbf{n} = \hat{\mathbf{n}}$,

$$\mathsf{A}(\mathbf{n}, \mathbf{n}) = - \sum_{\hat{\mathbf{n}} \in \Omega \setminus \{\mathbf{n}\}} \mathsf{A}(\mathbf{n}, \hat{\mathbf{n}}). \tag{16b}$$

Compute the stationary probability distribution $P(\mathbf{n})$ from (15), then calculate the system Key Performance Indicators (KPIs) using expressions in analytic form. Use the expression

$$N_s = \sum_{\mathbf{n} \in \Omega} P(\mathbf{n}) \, \mathbf{n} \cdot \mathbf{e}_s, \quad s \in \mathcal{S}, \tag{17}$$

to calculate the mean number of servicing requests at the $s^{\mathrm{th}}$ NSI, the expression

$$N = \sum_{\mathbf{n} \in \Omega} P(\mathbf{n}) \, \mathbf{n} \cdot \mathbf{j} \tag{18}$$

to calculate the mean number of servicing requests at the system, and the expression

$$\mathsf{P}\{\mathcal{A}\} = \sum_{\mathbf{n} \in \mathcal{A}} P(\mathbf{n}), \quad \mathcal{A} \subseteq \Omega, \tag{19}$$

to calculate the probability of an event $\mathcal{A}$ determined using the main sets (10)–(13). Let $P_s^{\mathrm{adm}}$ and $B_s^{\mathrm{block}}$ represent the probability of events $\Omega_s^{\mathrm{dad}} \cup \Omega_s^{\mathrm{vpad}}$ and $\Omega_s^{\mathrm{block}}$, respectively, i.e., the admission and blocking probabilities for arriving requests at the $s^{\mathrm{th}}$ NSI. Moreover, use the expression

$$\varphi_s = \frac{1}{Q_s} \sum_{\mathbf{n} \in \Omega} P(\mathbf{n}) \, (\mathbf{n} \odot \mathbf{b}) \cdot \mathbf{e}_s, \quad s \in \mathcal{S}, \tag{20}$$

to calculate the average utilization of the guaranteed network capacity of the $s^{\mathrm{th}}$ NSI, and the expression

$$\sigma = \frac{1}{C} \sum_{s=1}^{S} \sum_{\mathbf{n} \in \Omega} P(\mathbf{n}) \, (\mathbf{n} \odot \mathbf{b}) \cdot \mathbf{e}_s \tag{21}$$

to calculate the average utilization of the total network capacity of the 5G BS.

Given the above expressions, evaluate and analyze the efficiency of the proposed Pre-emption-based Prioritization (PP) scheme versus the classical Resource Reservation (RR) scheme in the following section.

## 4. Numerical Analysis

Perform a qualitative comparative analysis of the proposed Pre-emption-based Prioritization (PP) scheme versus the classical Resource Reservation (RR) scheme. Particularly

consider the Key Performance Indicators (KPIs) calculated using the expressions (17), (19), (20) and (21) in the previous section.

Consider the case example of three NSIs instantiated at one 5G BS. Dedicate the 1st NSI to 4K Live Video application instances. Next, dedicate the 2nd NSI to 4K 360-degree VR Panoramic Video application instances. Lastly, dedicate the 3rd NSI to 8K FOV VR Video application instances. Therefore, assign:

- The highest priority to servicing 4K Live Video requests at the 1st NSI;
- The medium priority to servicing 4K 360-degree VR Panoramic Video requests at the 2nd NSI;
- The lowest priority to servicing 8K FOV VR Video requests at the 3rd NSI.

Compare the performance when using both PP and RR-schemes versus the uniform offered load $\rho$ at the NSIs. Find all the input parameters in Table 2.

**Table 2.** Input parameters of comparative analysis [41].

| Parameter | Value RR-Scheme | Value PP-Scheme | Unit of Measure |
|:---:|:---:|:---:|:---:|
| $C$ | 5.0 | | Gbps |
| $Q_1, Q_2, Q_3$ | 1.0, 1.5, 2.5 | | Gbps |
| $C_1, C_2, C_3$ | $Q_1, Q_2, Q_3$ | 1.5, 2.0, 3.0 | Gbps |
| $b_1, b_2, b_3$ | 0.04, 0.08, 0.1 | | Gbps |
| $\rho$ | from 5 to 100 | | - |
| $\mu_1, \mu_2, \mu_3$ | 60, 30, 45 | | min$^{-1}$ |
| $\lambda_1, \lambda_2, \lambda_3$ | $\rho\mu_1, \rho\mu_2, \rho\mu_3$ | | requests/min |

Depictions of the obtained results for the considered KPIs are given in Figures 4–9, where

$$\Delta\,\mathrm{KPI} = |\mathrm{KPI}_{\text{PP-scheme}} - \mathrm{KPI}_{\text{RR-scheme}}|/\mathrm{KPI}_{\text{RR-scheme}} \tag{22}$$

represents the relative gain (https://mathworld.wolfram.com/PercentageError.html, accessed 1 September 2022) in terms of given KPI.

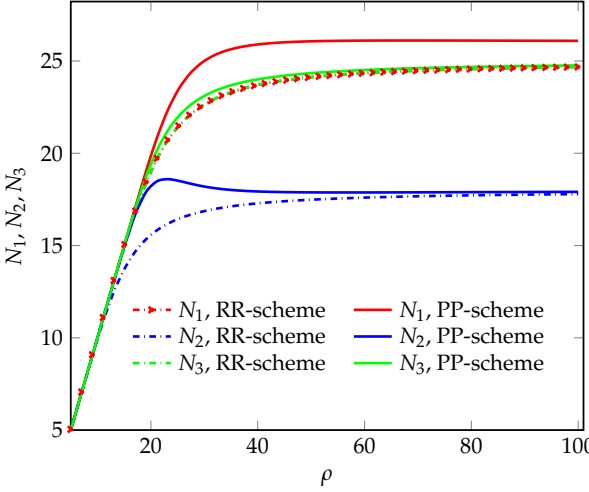

**Figure 4.** The mean numbers of servicing requests vs. the offered load.

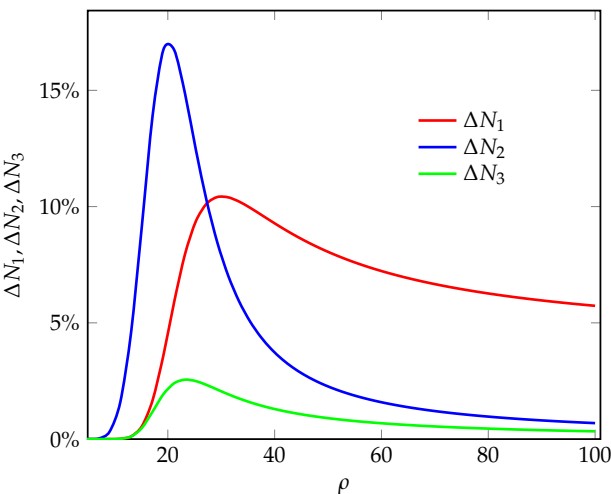

**Figure 5.** The percentage increase in the mean numbers of servicing requests vs. the offered load.

According to Figures 4 and 5, the mean numbers of servicing requests at the NSIs significantly increase with proposed PP-scheme compared to the classical RR-scheme. This result is the consequence of allowing violation of the guaranteed network capacities of the NSIs. With an increase up to approximately 16.75% when $\rho$ equals value 21.2, the 2nd NSI gains the most when using proposed PP-scheme given the input parameters in Table 2. As for the 1st NSI, instantiated to service 4K Live Video requests with "low" requirements, it gains up to approximately 10.49% when $\rho$ equals value 29.29. As for the 3rd NSI, instantiated to service 8K FOV VR video requests with "high" requirements, it gains up to approximately 2.6% when $\rho$ equals value 23.22. Thus, compared to the classical RR-scheme, the proposed PP-scheme always provides better performance in terms of mean numbers of servicing requests at the NSIs.

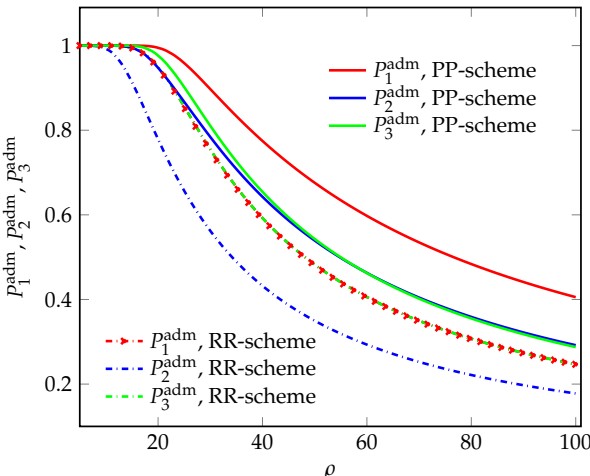

**Figure 6.** The admission probabilities for arriving requests vs. the offered load.

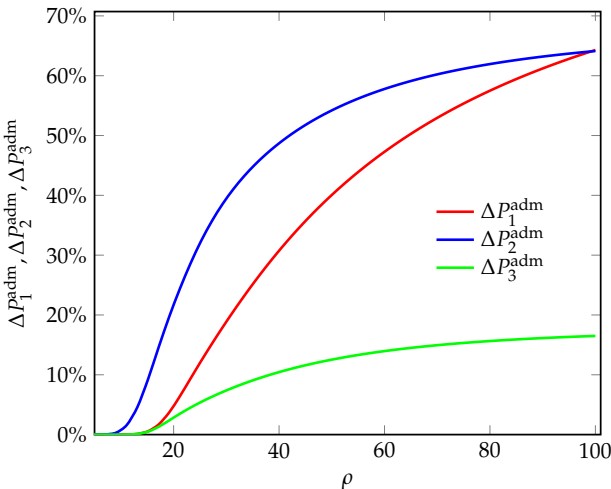

**Figure 7.** The percentage increase in the admission probabilities for arriving requests vs. the offered load.

In Figures 6 and 7, the admission probabilities are greatly improved with the proposed PP-scheme. Consequently, the blocking probabilities greatly decrease (Figures 8 and 9). Thus, the proposed PP-scheme also provides better performance, in terms of admission and blocking probabilities for arriving requests at the NSIs.

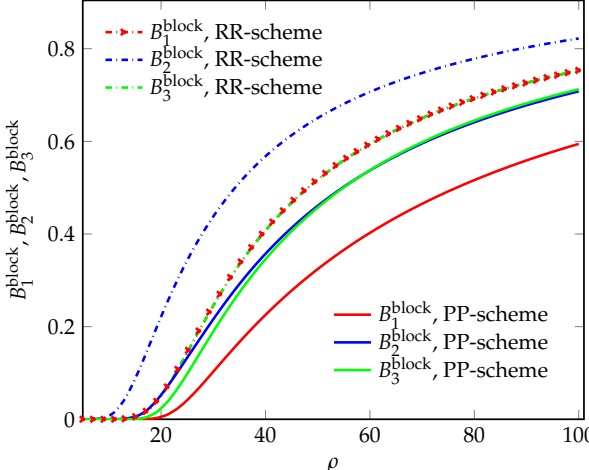

**Figure 8.** The blocking probabilities for arriving requests vs. the offered load.

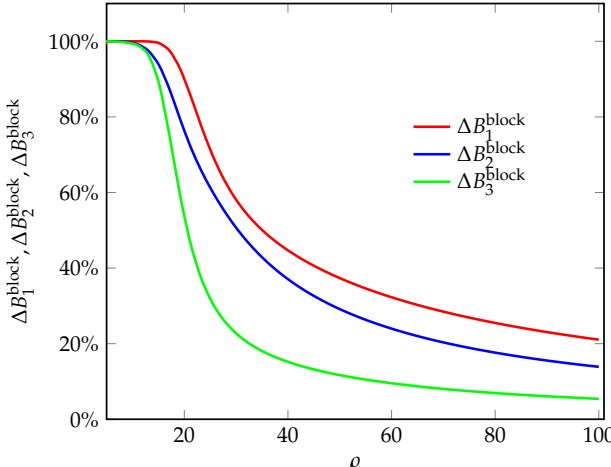

**Figure 9.** The percentage decrease in the blocking probabilities for arriving requests vs. the offered load.

In Figure 10, the average utilization of the guaranteed network capacities improved when using the proposed PP-scheme. This is especially evident for the 2nd NSI in Figures 5 and 7. Indeed, by flexibly using available parts of neighboring guaranteed network capacities of the 1st and/or 3rd NSIs, the 2nd NSI can service more 4K 360-degree VR Panoramic Video requests at middle workload values, i.e., when arriving requests can only be admitted for service with their respective guaranteed network capacities. One can also note that due to the low requirements for the throughput $b_1$, with a simultaneous increase in the load $\rho$, the requests of the 1st NSI always manage to occupy the underused capacities of other NSIs, that is why $\phi_1$ exceeds unity and lies above the rest of the curves.

In Figure 11 finally, the average utilization of the total network capacities improved by up to 10% at medium and high workload when using the proposed PP-scheme, compared to the classical RR-scheme. Note that, unlike the average utilization of the guaranteed network capacities (Figure 10), the average utilization of the total network capacities cannot exceed 100%.

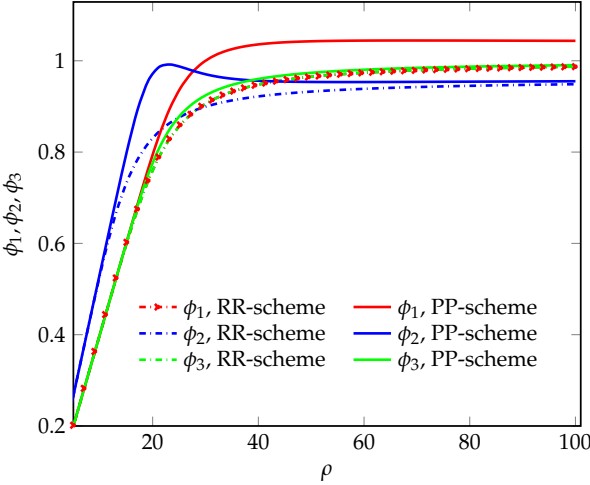

**Figure 10.** The average utilization of the guaranteed network capacities vs. the offered load.

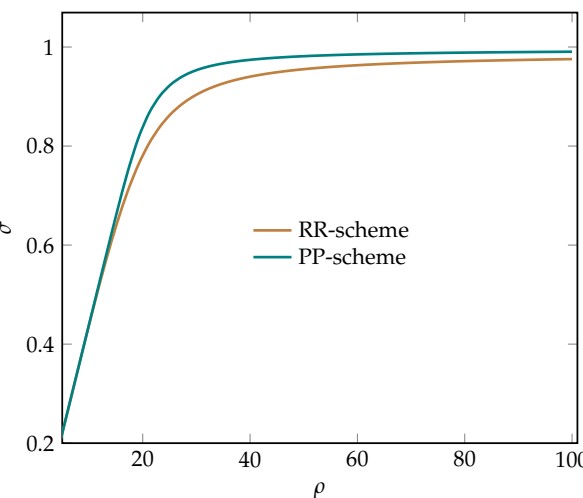

**Figure 11.** The average utilization of the total network capacities vs. the offered load.

To conclude, the numerical analysis was conducted given simultaneously increased uniform offered loads at the NSIs, while the average service times and service requirements were individual. The given Queueing System (QS) model in Section 3 facilitates the evaluation or estimation of proposed PP-scheme efficiency with respect to some baseline. As one key result, the 2nd NSI, instantiated to service requests with "medium" requirements, significantly gains in performance when using the proposed PP-scheme, compared to the classical RR-scheme. Indeed, the arriving 4K 360-degree VR Panoramic Video requests at the 2nd NSI use more often the guaranteed network capacities of other neighboring NSIs. The highlighted result is illustrated in the figures in this section.

## 5. Conclusions

This paper proposes a Pre-emption-based Prioritization (PP) scheme "merging" the classical Resource Reservation (RR) and Service Prioritization schemes. A Queueing System (QS) model analyzing the operation of multiple Network Slice Instances (NSIs) at common 5G BSs is provided to evaluate or estimate the proposed PP-scheme efficiency. Expressions are given to calculate system Key Performance Indicators (KPIs): the mean numbers of servicing requests, the admission and blocking probabilities for arriving requests, and the average utilization of the total and guaranteed network capacities. A qualitative numerical analysis comparing the proposed PP-scheme with the classical RR-scheme is given. Thus, the operation of three NSIs instantiated for 4K Live Video, 4K 360-degree VR Panoramic Video and 8K FOV VR Video requests is considered. The key results of the conducted comparative analysis are the following. Given some baseline, the proposed PP-scheme can provide:

- Up to more than 60% gain in terms of admission probability of arriving 4K 360-degree VR Panoramic Video requests at the 2nd NSI;
- Up to 100% gain in terms of blocking probabilities of arriving requests;
- Up to 15% in terms of average utilization of the guaranteed network capacity of the 2nd NSI.

As next stage of this research topic, the PP-scheme efficiency can be evaluated or estimated given a QS model supporting also data applications using Transmission Control Protocol (TCP), e.g., Web Browsing, File Download, Social Networking, etc. Future research can investigate the optimal capacity settings for all NSIs at common 5G BSs given the PP-scheme.

**Author Contributions:** Conceptualization, Y.A., E.M. and Y.G.; Data curation, Y.A.; Formal analysis, Y.A.; Funding acquisition, E.M.; Investigation, Y.A.; Methodology, Y.A., E.M. and Y.G.; Project administration, E.M. and Y.G.; Resources, Y.A.; Software, Y.A.; Supervision, E.M. and Y.G.; Validation,

Y.A.; Visualization, Y.A. and E.M.; Writing—original draft, Y.A.; Writing—review and editing, Y.A., E.M. and Y.G. All authors have read and agreed to the published version of the manuscript.

**Funding:** The research was funded by the Russian Science Foundation, project no.22-79-10053, (https://rscf.ru/en/project/22-79-10053/, accessed 1 September 2022).

**Institutional Review Board Statement:** Not applicable.

**Informed Consent Statement:** Not applicable.

**Data Availability Statement:** Not applicable.

**Conflicts of Interest:** The authors declare no conflicts of interest.

## Abbreviations

The following abbreviations are used in this manuscript:

| | |
|---|---|
| 3GPP | Third Generation Partnership Project |
| 5G | Fifth generation |
| BE | Best effort |
| BG | Best effort with minimum guaranteed |
| BS | Base station |
| CN | Core network |
| DN | Data network |
| E2E | End-to-End |
| FOV | Field of vision/view |
| GB | Guaranteed bit rate |
| GSM | Groupe Speciale Mobile |
| IoT | Internet of Things |
| IoV | Internet of Vehicles |
| MNO | Mobile network operator |
| NS | Network slicing |
| NSI | Network slice instance |
| PP | Pre-emption-based prioritization |
| QoS | Quality of Service |
| QS | Queueing system |
| RAC | Radio admission control |
| RAT | Radio access technology |
| RR | Resource reservation |
| TN | Transport network |
| VoIP | Voice over Internet Protocol |
| VR | Virtual reality |

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
