# Peer review of "Modeling and Analyzing Preemption-Based Service Prioritization in 5G Networks Slicing Framework"

_futureinternet, doi:10.3390/fi14100299_

Round 1

Reviewer 1 Report

In the paper, the authors propose a queuing system model to analyze the operation of a 5G base station that accommodates multiple Network Slice Instances. They describe the radio admission control in terms of blocking, direct admission or via preemption admission. The queueing model is analyzed in the steady-state, and the steady state probabilities are obtained numerically to calculate the main performance metrics.

It should be noted that there are no serious comments on the paper, the text is well-organized and the most explanations are clear to a reader. The paper clearly fits the journal’s scope. Thus, the paper is recommended for publishing in Future Internet Journal.

Reviewer 2 Report

The paper considers the problem of Network Slice Instances (NSIs) management in 5G systems. Authors consider an example of admission control and preemption policy and build the corresponding mathematical model.

The reviewer has the following major comments

1. System model

In Section 2, authors assume that all data flows are characterized by three variables: (i) the average arrival rate $\lambda$, (ii) the average bitrate $b$, and (iii) the average service time $\mu^{-1}$. While this model is adequate for real-time applications, such as voice and real-time video, it is not adequate for the data applications that use TCP as an underlying transport layer protocol. For example, when a user downloads a web page of a given size $S$, the corresponding TCP flow uses all available (free) capacity $C$. Thus, the flow service time is $S/C$. When two TCP flows share the same link/path they obtain equal share of the link/path capacity, and therefore the service time of each flow doubles.

Note that the considered by authors flow model ignores that the service time of the data flow depends on the available capacity and the number of concurrent flows that share the same resource pool. Thus, it can not be used for modeling data applications such as Web browsing, File download, Social networking, and others that use TCP. Note that, in the experiments, authors consider exactly TCP-based data flows.

2. Admission control policy

First, authors consider a very specific admission control and preemption policy. While authors admit that other policies can be used, they do not compare the proposed policy with the existing ones (at least some of them) numerically.     

Second, authors apply admission control to TCP-based data flows. However, in real systems TCP flows are typically not dropped. When the number of flows increases they share a common resource pool, each flow obtains proportionally lower data rate and service time of _all_ flows increases. Only when the service time of the TCP flow exceeds some threshold, the user itselfs drops the corresponding flow. 

Taking into account comments (1) and (2), the reviewer has the following suggestions. Authors should either (i) consider a more adequate data flow model that takes into account the mentioned above peculiarities of TCP data flows and consequently reconsider their mathematical model, or (ii) consider only scenarios with real-time applications that for sure will limit the applicability area of the considered solution.

3. Numerical results

In Section 4, authors provide numerical results obtained based on their mathematical model. However, authors: (i) do not validate the developed model (e.g., with simulations), (ii) do not provide any clear conclusions from the obtained results. The presented plots only show some numbers that increase/decrease with the arrival rate. But, what does that mean for practice? Whether the proposed policy is "good" or "bad" and in what sense?

Also, the numerical results lack comparison with the existing admission control policies (see comment 2).   

4. Quality of presentation

The paper requires careful proofread. Examples of improper language usage are: “consider the next scenario”, “is approaching Q1/b1 from 1”, “which may seem to be an inflection point”, “with the next fact”, “the more requests may be servicing at the system”, “In Figure 6 observe the impact on”, etc.

Reviewer 3 Report

From the review of this paper I have come to the following conclusions:

1- I suggest that the title of the paper rewritten to be general 

2-  The abstract should contain any conclusion that attracted readers’ attention.

3- in introduction part ,  Authors should focus on the (practical) interest of their investigation. Avoid the use of “see (1-4)”.

4- clarification on the application of the problem studied is needed.

5- Would you please cite all equations that you did not derive?

6-  Can you Provide the quantitative analysis of the obtained results in the abstract section with parameter range.

7- Abbreviations section ,  should contain all the abbreviations are used in this manuscript

Round 2

Reviewer 2 Report

While the technical content of the paper has been significantly improved, the following issues should be clarified/elaborated.

In section 4, authors consider 3 services: Video call, Audio streaming, VoIP. 

First, it is not clear what is the difference between  Audio streaming and VoIP. If Audio streaming is the listening of the music from some service (e.g., Apple Music) it is still delivered via TCP and, thus, do not correspond to the model.

Second, the bitrates of  26.26, 28.68, 9.56 Mbps seem to be inadequate to the existing applications. A VoIP flow does not consume 9.56 Mbps. Video call has much higher bitrate than the audio/VoIP call. The same question is releted to the values of parameter \mu.

The text still needs improvement. Examples are as follows:

- “2nd — to Audio streaming services” verb is missing. The usage of long dash does not follow English grammar

- “1 -> 100” is very strange way to describe the range of values

- “\ro = +-18” what does that mean?

- “In any case, here the proposed solution NS-model wins” too informal

- “the proposed solution NS-model”. The solution naming is not clear. As far as the reviewer can understand from the paper, the solution is the preemption mechanism (given by eq (7)), while the mathematical model (given in Section 3) is only used to analyze its efficiency. So, the solution cannot be based on the model.

Conclusion should be extended with specific numbers (gains) obtained in the paper, e.g. “the proposed method provides up to XX% gain in terms of the given KPI with respect to some baseline”
